# Excess maternal mortality in Brazil: Regional inequalities and trajectories during the COVID-19 epidemic

Jesem Orellana[1]*, Nadège Jacques[2], Daniel Gray Paschoal Leventhal[3], Lihsieh Marrero[4], Lina Sofía Morón-Duarte[5,6]

1 Leônidas and Maria Deane Institute, Oswaldo Cruz Foundation, Manaus, Amazonas, Brazil, 2 Postgraduate Program in Epidemiology, Department of Social Medicine, Federal University of Pelotas, Pelotas, Rio Grande do Sul, Brazil, 3 Independent Researcher, Takoma Park, Maryland, United States of America, 4 Department of Nursing, Amazonas State University, Manaus, Amazonas, Brazil, 5 Global Institute of Clinical Excellence, Keralty, Bogotá, Distrito Capital, Colombia, 6 Translational Research Group, Sanitas University Foundation, Bogotá, Distrito Capital, Colombia

* jesem.orellana@gmail.com

**Data Availability Statement:** The data underlying the results presented in the study are available from https://datasus.saude.gov.br/informacoes-de-saude-tabnet/.

## Abstract

### Background

The COVID-19 pandemic has exceeded 6 million known disease-related deaths and there is evidence of an increase in maternal deaths, especially in low- and middle-income countries. We aimed to estimate excess maternal deaths in Brazil and its macroregions as well as their trajectories in the first 15 months of the COVID-19 epidemic.

### Methods

This study evaluated maternal deaths from the Mortality Information System of the Ministry of Health, with excess deaths being assessed between March 2020 and May 2021 by quasi-Poisson generalized additive models adjusted for overdispersion. Observed deaths were compared to deaths expected without the pandemic, accompanied by 95% confidence intervals according to region, age group, and trimester of occurrence. Analyses were conducted in R version 3.6.1 and RStudio version 1.2.1335.

### Results

There were 3,291 notified maternal deaths during the study period, resulting in a 70% excess of deaths regardless of region, while in the North, Northeast, South and Southeast regions, excess deaths occurred regardless of age group. Excess deaths occurred in the March-May 2021 trimester regardless of region and age group. Excess deaths were observed in the Southeast region for the 25-36-year-old age group regardless of the trimester assessed, and in the North, Central-West and South regions, the only period in which excess deaths were not observed was September-November 2020. Excess deaths regardless of trimester were observed in the 37-49-year-old age group in the North region, and the South region displayed explosive behavior from March-May 2021, with a 375% excess of deaths.

**Funding:** The authors gratefully acknowledge support from the Fundação de Amparo à Pesquisa do Estado do Amazonas and Fundação Oswaldo Cruz, (project grant 04/2022/FIOCRUZ/FAPEAM/FAPERO) – Programa de Apoio à Pesquisa de Inovação Tecnológica – Inovação na Amazônia, awarded to JO and LM.

**Competing interests:** The authors have declared that no competing interests exist.

## Conclusions

Excess maternal deaths, with geographically heterogenous trajectories and consistently high patterns at the time of the epidemic's greatest impact, reflect not only the previous effect of socioeconomic inequalities and of limited access to maternal health services, but most of all the precarious management of Brazil's health crisis.

## Introduction

Maternal mortality is a public health problem related to social inequality, access to health services, and limited organisation and quality of obstetric care, particularly in low- and middle-income countries (LMICs) [1, 2]. Not coincidentally, global reduction of the maternal mortality ratio is one of the priority goals of the 2030 Agenda for Sustainable Development [3].

Beyond the more than 6 million confirmed deaths by COVID-19 [4], the pandemic has devastated health systems, the global economy, and society at large [5]. Evidence suggests that the global pattern of maternal mortality has increased significantly during this time [6, 7].

Brazil has implemented human and reproductive rights policies and measures to reduce maternal mortality since the 1980s [8, 9]. Between 1990 and 1999, the number of maternal deaths in the country fell significantly, followed by a relatively stable pattern between 2000 and 2013, then a slight drop between 2017 and 2019 [10]. However, recent studies suggest a reversal in the decreasing pattern of maternal deaths [11–13] because of serious failings in the COVID-19 epidemic's management in Brazil [14, 15].

There are few studies that quantify excess maternal deaths in the COVID-19 pandemic [16], especially during the period of greatest health impact and in LMICs, where adequate surveillance and notification of maternal deaths are a historical challenge [16, 17]. Using the counterfactual approach, this study aimed to estimate excess maternal deaths in Brazil and their trajectories in the first 15 months of the COVID-19 epidemic.

## Materials and methods

### Study type, data sources, and units of analysis

This exploratory mixed ecologic study [18] employed the counterfactual approach [19] using data from death certificates in the Mortality Information System (SIM) of the Ministry of Health of Brazil during the period of March 2015 to May 2021. Due to Brazil's size, inequalities in access to health services, and heterogeneous pattern of the epidemic [20], the country's five macroregions (Southeast, Northeast, North, Central-West, and South) were included as units of analysis. The North and Northeast regions are historically less socioeconomically developed than the Southeast and South regions, which harbour not only the country's largest populations but also its largest industrial and agro-export capacity [21].

Data from 2015 to 2020 [22] are finalized. Those from 2021 were made available preliminarily, up to December 2021. The State Health Departments are given 60 days following the end of the month of each death's occurrence to transfer their data to SIM's national platform, though this could possibly take longer [23]. To guarantee reliable coverage to the partial 2021 analyses, deaths occurring until May 2021 were included. Therefore, our analyses refer to a period greater than 360 days following their occurrence with respect to the date the May 2021 deaths data were made available preliminarily, reducing the possibility of a delay in notification [24].

## Operational definitions

Maternal deaths were selected according to criteria adopted by the Brazilian Ministry of Health's protocol of special codes for mortality, which includes maternal mortality in women 10–49 years old (adopted for this study to estimate excess deaths) considering the victim's place of residence. Maternal death is death during gestation or up to 42 days after the termination of gestation, regardless of the duration or location of the pregnancy and due to any cause related to or aggravated by the pregnancy or related measures, but not to accidental or incidental causes [25].

The following codes of the International Statistical Classification of Diseases and Related Health Problems [26] were considered: O00-O99 (Pregnancy, childbirth and the puerperium), except deaths outside of the 42-day postpartum period (codes O96 and O97); B20-B24 (Human immunodeficiency virus [HIV] disease); D39.2 (malignant or invasive hydatidiform mole); or E23.0 (Hypopituitarism), with a gestational report at the time of death or up to 42 days before death. Also included are codes M83.0 (Puerperal osteomalacia), A34 (Obstetrical tetanus), or F53 (Mental and behavioural disorders associated with the puerperium, not elsewhere classified), with death having occurred up to 42 days after termination of pregnancy or when there was no information on the time elapsed between the end of pregnancy and death. In case of inconsistencies between the declared maternal cause and the time of death (during the pregnancy, delivery, or miscarriage/abortion, during the postpartum period up to 42 days, during the postpartum period, between 43 days and 1 year or outside these periods), information on the basic cause of death was prioritised [25]. According to the World Health Organization (WHO), excess deaths represent the number of deaths situated above an expected value, following a previously observed mortality pattern in the population [27].

## Data analysis

Data from March 2020 to May 2021 were analysed. This period was subdivided into five trimesters: March-May 2020, June-August 2020, September-November 2020, December 2020-February 2021, and March-May 2021. The observed mortality data from SIM (March 2015-February 2020) were used to estimate the expected deaths in each of the five trimesters assessed.

The counterfactual approach to assessing excess mortality has proven useful for estimating the indirect effects of crises in which health services are severely compromised and excess deaths are concentrated in socioeconomically disadvantaged groups. Therefore, the overall effects of the COVID-19 pandemic can be better understood through this approach, which reasonably represents the differences between observed deaths and the number of deaths that would have occurred in the absence of the pandemic [19]. Estimates of expected deaths from March 2020 to May 2021 were obtained through *quasipoisson* generalized additive models [28] corrected for overdispersion. The variables trimester, age group, and the interaction between year of occurrence with age group were considered as predictors in each regional model and at the country-level with p values <20% assumed as statistically significant. Year of occurrence was adjusted non-parametrically (*spline*) to capture possible non-linear mortality trends in the period assessed. We estimated the number of expected deaths using the adjusted models. Due to the reduced number of maternal deaths that occurred before the age of 20 and the reduced accuracy in our maternal death estimates, age groups were aggregated in the following brackets: 10–24 years, 25–36 years, and 37–49 years.

Estimates of excess maternal mortality were based on the calculated ratio between the number of observed maternal deaths and those expected if the pandemic did not occur for each country region and at the national level, stratified by each age group and trimester analysed

[19]. The width of the 95% confidence intervals of each one of the expected point estimates was compared to the corresponding observed value to assess statistically significant differences between observed and expected values. According to the WHO, expressing deaths in excess through a percentage enables decision-makers to understand [27], so our results were presented in that manner. Analyses were performed in R version 3.6.1 and RStudio version 1.2.1335 (http://www.r-project.org).

### Ethical considerations

This study did not need approval from an Ethics Committee on Research on Human Beings due to the use of de-identified and publicly available data, as recommended by Resolution No. 510/2016 of the National Health Council.

## Results

There were 3,291 analysed maternal deaths in Brazil between March 2020 and May 2021, with 1,179, 983, 497, 318, and 314 in the Southeast, Northeast, North, South, and Central-West regions, respectively. A 70% excess in maternal deaths was observed in all Brazil, with the largest excess, 88%, observed in the 25-36-year-old age group (Table 1).

The analysis stratified by region displayed excess maternal mortality regardless of region, with values of 78%, 67%, 75%, 89%, and 63% in the North, Northeast, Central-West, South, and Southeast regions, respectively (Table 1). Furthermore, all age groups exhibited excess maternal mortality in the North, Northeast, Southeast and South regions. In the Central-West region, observed and expected values were compatible for maternal deaths in the 10-24-year-old age group (Table 1).

In the North and Northeast (Fig 1), excess maternal mortality in the 10-24-year-old age group was unclear across all trimesters assessed. In the Central-West and South regions, the 10-24-year-old age group displayed excess deaths in the December/20-February/2021 and March-May 2021 trimesters (Fig 2).

In the 25-36-year-old age group, excess maternal mortality was observed in all evaluated trimesters except September-November 2020 in the North (Fig 1), Central-West and South (Fig 2) regions. Excess maternal mortality was not observed among women in the North (Fig 1) Central-West and South regions (Fig 2) in one of the five trimesters assessed. On the other hand, excess maternal mortality occurred among women 25–36 years old in the Northeast (Fig 1) and Southeast region (Fig 3) regardless of trimester.

For the 37-49-year-old age group, excess maternal deaths were observed regardless of trimester assessed in the North region (Fig 1). In the Northeast region, excess maternal deaths occurred in all trimesters except December 2020-February 2021 (Fig 1). In the Brazilian South region, excess maternal mortality was present in three of the five evaluated trimesters, and reached 375% (38/8) in the March-May 2021 trimester (Fig 2). Excess maternal deaths were only observed in the March-May 2021 trimester in the Southeast region (Fig 3). In all regions and regardless of age group, excess maternal deaths were observed in the March-May 2021 trimester.

The analysis of the interaction term was statistically significant at the country level and in all regional models (P value <20%), except in the Central-West region (Table 2).

Around 80% of the causes of death during the study period pertained to complications predominantly related to the puerperium and other obstetric conditions, in addition to oedema and proteinuria and hypertensive disorders in pregnancy, childbirth, and the puerperium not elsewhere classified (S1 Table), with substantial variations in the main causes of death over the

**Table 1. Percentage of excess maternal deaths by age group and region, Brazil, 2020–2021.**

| Variables | Observed | Expected | Expected* | Mortality ratio |
|---|---|---|---|---|
| | (n) | (n) | (CI95%) | (%) |
| North | | | | |
| **Age group (years)** | | | | |
| 10–24 | **151** | 110 | 89–134 | **37%** |
| 25–36 | **267** | 136 | 100–170 | **96%** |
| 37–49 | **79** | 34 | 24–48 | **132%** |
| Overall | **497** | 280 | 213–352 | **78%** |
| Northeast | | | | |
| **Age group (years)** | | | | |
| 10–24 | **227** | 175 | 149–202 | **30%** |
| 25–36 | **552** | 309 | 269–345 | **79%** |
| 37–49 | **204** | 104 | 76–135 | **96%** |
| Overall | **983** | 588 | 494–682 | **67%** |
| Central-West | | | | |
| **Age group (years)** | | | | |
| 10–24 | 61 | 53 | 41–67 | 15% |
| 25–36 | **174** | 96 | 71–119 | **81%** |
| 37–49 | **79** | 30 | 22–40 | **163%** |
| Overall | **314** | 179 | 134–226 | **75%** |
| South | | | | |
| **Age group (years)** | | | | |
| 10–24 | 69 | 49 | 29–68 | **41%** |
| 25–36 | **171** | 80 | 61–98 | **114%** |
| 37–49 | **78** | 39 | 27–51 | **100%** |
| Overall | **318** | 168 | 117–217 | **89%** |
| Southeast | | | | |
| **Age group (years)** | | | | |
| 10–24 | **248** | 173 | 146–203 | **43%** |
| 25–36 | **678** | 361 | 305–417 | **88%** |
| 37–49 | **253** | 189 | 158–220 | **34%** |
| Overall | **1,179** | 723 | 609–840 | **63%** |
| Brazil | | | | |
| **Age group (years)** | | | | |
| 10–24 | **756** | 560 | 454–674 | **35%** |
| 25–36 | **1,842** | 982 | 806–1,149 | **88%** |
| 37–49 | **693** | 396 | 307–494 | **75%** |
| Overall | **3,291** | 1,938 | 1,567–2,317 | **70%** |

**Source:** SIM/DATASUS.

*Confidence interval (CI) estimate of the number of expected maternal deaths. **Bold**: observed value above the upper bound of the confidence interval estimate of the expected value.

evaluated trimesters, including viral diseases complicating pregnancy, severe pre-eclampsia and unspecified pre-eclampsia, for example (S2 Table).

## Discussion

Our study showed a 70% excess in maternal deaths in Brazil during the COVID-19 pandemic, but with geographically heterogeneous trajectories and a substantially high pattern in the

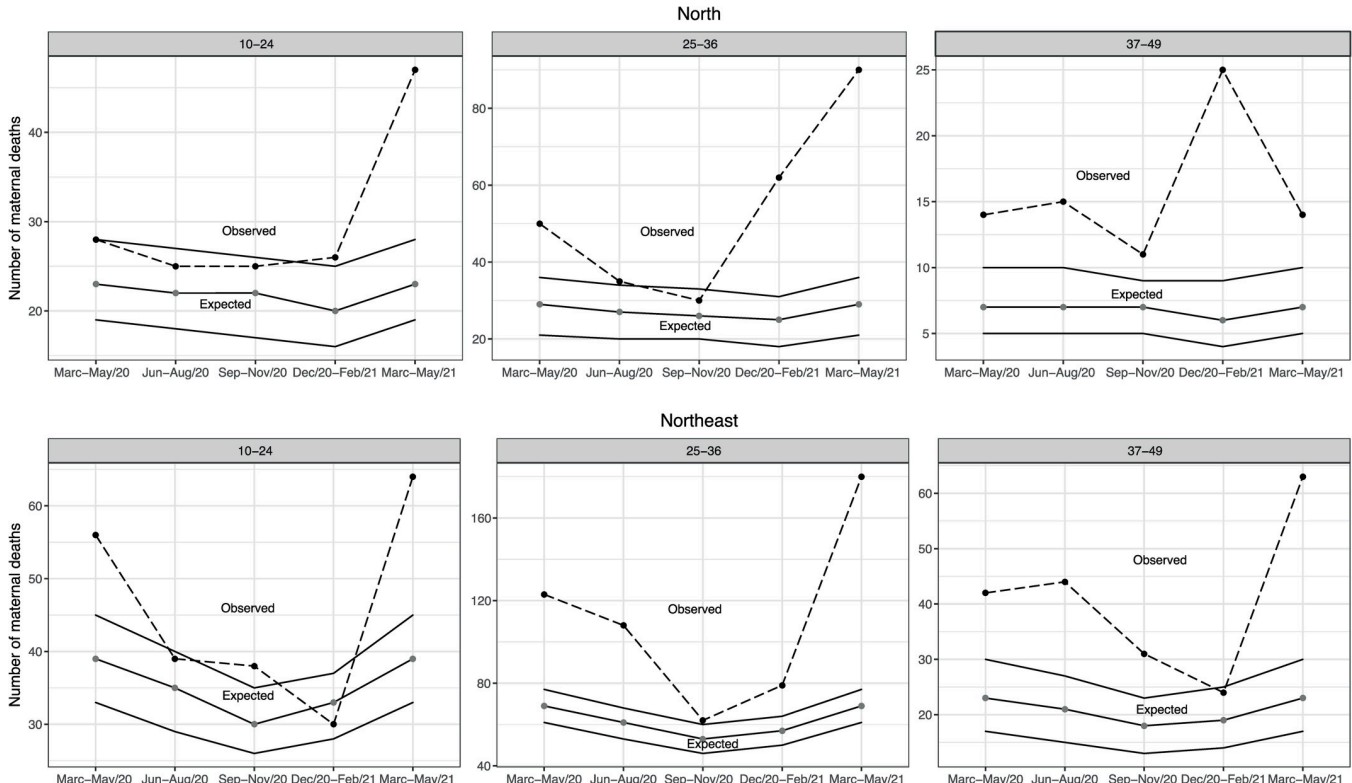

**Fig 1.** Observed and expected maternal deaths according to age group and five consecutive trimonthly periods, North and Northeast regions, Brazil 2020–2021.

March-May 2021 trimester, regardless of age group and region. The elevated burden of maternal deaths in the COVID-19 epidemic suggests that timely access to healthcare services and the quality of services provided have deteriorated, especially at the time of the epidemic's greatest impact.

More than two years after notification of the first case of COVID-19, Brazil remains among countries most seriously affected by the direct effects of the pandemic [29], with close to 677,000 confirmed deaths by the disease, or 11% of total deaths globally [4]. The pandemic's indirect effects on other causes of mortality have also been observed in the country [20, 30] and the general pattern of excess maternal mortality reinforces the epidemic's dramatic development, compromising Brazil's efforts to reach Sustainable Development Goal (SDG) 3 by reducing maternal mortality and guaranteeing universal access to sexual and reproductive health for women by 2030 [3].

Flaws in early diagnosis and management of complications during pregnancy and childbirth were among the main factors associated with premature births, foetal deaths, low birth weight, and increased maternal mortality during the COVID-19 pandemic. Pre-eclampsia and eclampsia cause 7–15% of maternal deaths [31]. A systematic review with metanalysis suggests that COVID-19 can be associated with an elevated risk of pre-eclampsia, premature birth, and other adverse outcomes during pregnancy [32]. Studies from the United States, the United Kingdom, India, and Mexico have shown an increase in maternal mortality because of the COVID-19 pandemic [33].

In Brazil, vulnerabilities in obstetric care including difficulties accessing antenatal and specialized healthcare services and a dearth of qualified professionals to handle complications in

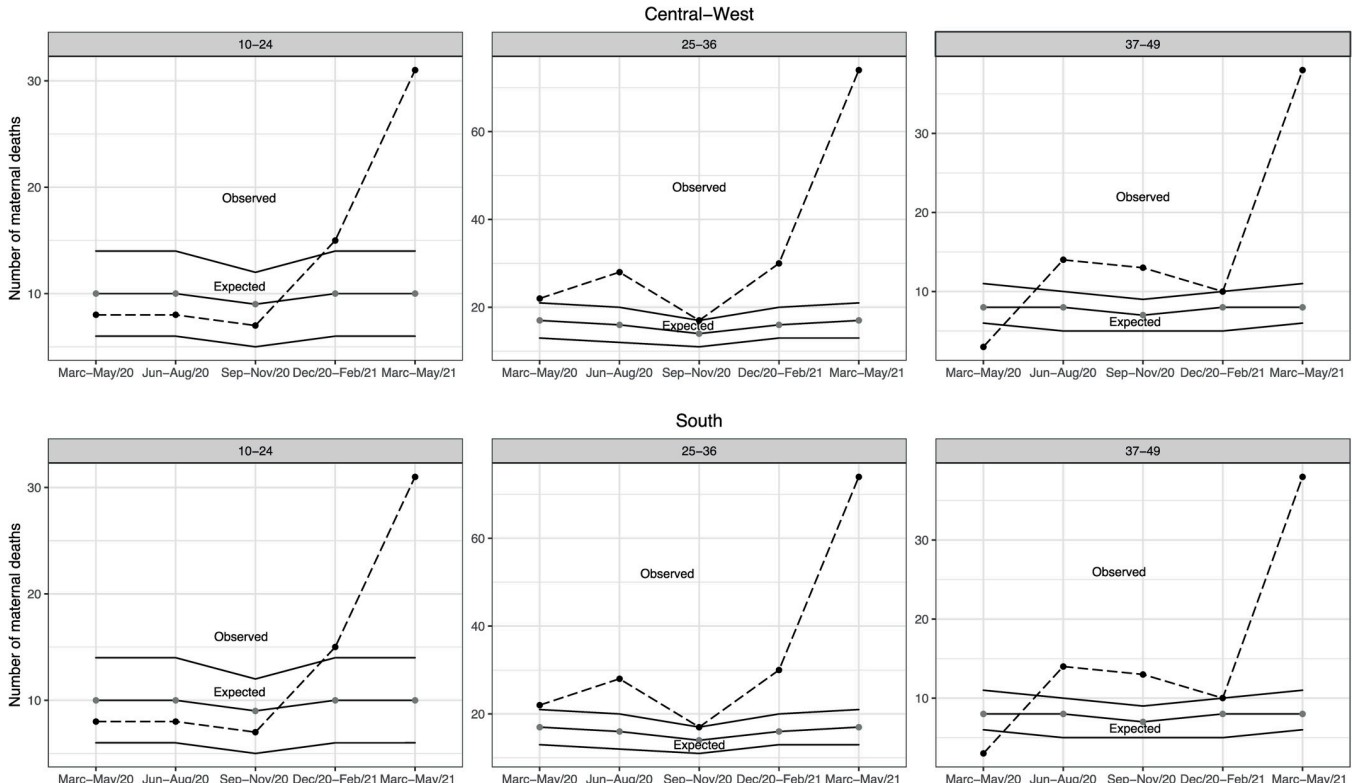

**Fig 2. Observed and expected maternal deaths according to age group and five consecutive trimonthly periods, Central-West and South Regions, Brazil 2020–2021.**

pregnancy, childbirth, and the postpartum period worsened during the COVID-19 pandemic [34]. The overwhelming burden on health services in the country compromised antenatal care, contributing to the increase in maternal deaths by causes that could have been avoided through adequate pregnancy care, such as pre-eclampsia and eclampsia. It can be observed that the increase in deaths by these causes coincided with the peaks of COVID-19 in the country (S2 Table). In general, the overloaded medical-hospital network (especially in the North and Northeast regions) [35], in addition to patients' reluctance to go to health services during the pandemic due to fear of contamination by SARS-CoV-2, the increasing number of confirmed COVID-19 new cases among healthcare professionals, a lack of personal protective equipment, limited precautions against infection, or even premature deaths among healthcare professionals [15, 36] may have diminished the already limited provision and utilization of maternal healthcare services.

As observed in different settings, the COVID-19 pandemic and its spatial-temporal effects are distinct within and between countries [13, 37]. Our results demonstrated substantial heterogeneity in excess maternal deaths among the different regions of Brazil, especially in the North, where this indicator was elevated in almost all trimesters assessed except in the 10-24-year-old age group.

Also noteworthy was the sustained excess of deaths (greater than 30%) in the March-May 2020 and June-August 2020 trimesters exclusively in the North and Northeast regions. This differed considerably from the patterns observed in the Southeast, South, and Central-West regions, which before the COVID-19 pandemic already had the largest supply of general and intensive care unit (ICU) hospital beds [37].

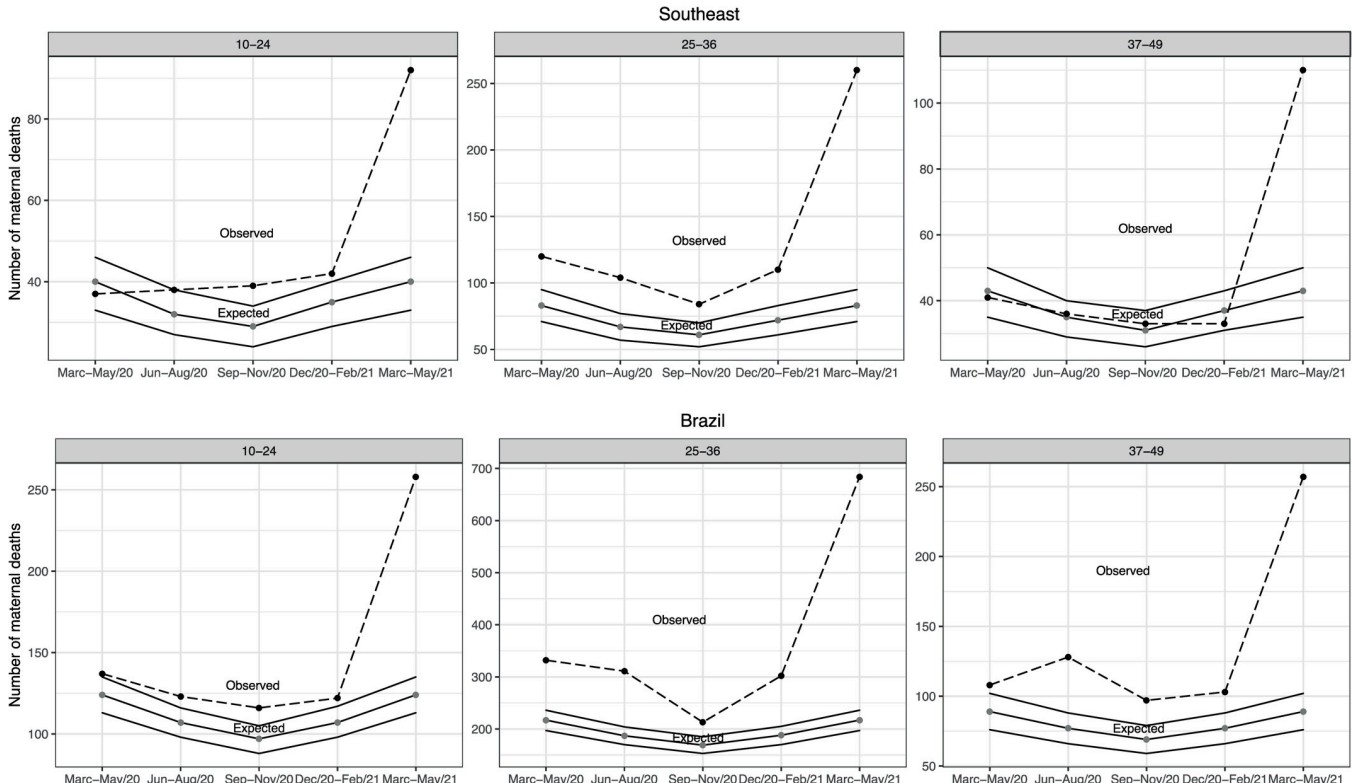

**Fig 3. Observed and expected maternal deaths according to age group and five consecutive trimonthly periods, Southeast region and Brazil, 2020–2021.**

During the first months of 2020, the effects of the COVID-19 pandemic in pregnant women and new-borns were unclear. However, gradually, studies began to show the impact of the disease on deaths among pregnant and postpartum women. A report from MBRRACE-UK showed that six in every ten women that died during or after pregnancy by COVID-19 or its complications belong to Black or minority ethnic groups [38].

Part of the differences found in excess maternal mortality in our study may be explained by the striking socioeconomic inequalities in the North and Northeast regions of Brazil [39, 40] which have the worst infrastructure and access to health services in the country [41]. The North region, most harshly punished across all trimesters assessed, was not as prepared to face the epidemic since, proportionally, it possessed the lowest number of ICU beds, physicians, and ventilators, an essential device for facing COVID-19 complications [42], of all macroregions. However, the number of available hospital beds by itself is a limited measure of the true ability and quality of healthcare services, as the density and distribution of healthcare professionals, the quality of hospital equipment, and the accessibility of health services appear to be more comprehensive indicators of this scenario [43].

The Northeast region, which represents close to 27% of the Brazilian population and contributes to almost half (47.9%) of all the estimated poverty in the country [44], was also severely impacted in the COVID-19 pandemic, not only with respect to the number of cases and of deaths, but also due to the aggravation of poverty [43, 45] and of socioeconomic and racial/ethnic inequalities [46]. Therefore, it is possible that the impacts of the COVID-19 epidemic have been greatest in precisely the most socioeconomically vulnerable regions of the country [47], and this is reflected in the unequal indicators of excess mortality.

**Table 2. Description of the interaction terms between the year of occurrence of death and age group at the regional and national levels, Brazil, 2020–2021.**

| North | |
|---|---|
| **Age group (years)** | **p-value** |
| 10–24 | 0.950 |
| 25–36 | **0.162** |
| 37–49 | 0.911 |

| Northeast | |
|---|---|
| **Age group (years)** | **p-value** |
| 10–24 | **0.001** |
| 25–36 | **0.060** |
| 37–49 | 0.154 |

| Central-West | |
|---|---|
| **Age group (years)** | **p-value** |
| 10–24 | 1.000 |
| 25–36 | 0.459 |
| 37–49 | 0.625 |

| South | |
|---|---|
| **Age group (years)** | **p-value** |
| 10–24 | **0.163** |
| 25–36 | **0.018** |
| 37–49 | 0.909 |

| Southeast | |
|---|---|
| **Age group (years)** | **p-value** |
| 10–24 | **0.052** |
| 25–36 | **0.173** |
| 37–49 | 0.349 |

| Brazil | |
|---|---|
| **Age group (years)** | **p-value** |
| 10–24 | **0.001** |
| 25–36 | **0.039** |
| 37–49 | **0.142** |

Despite the consistently strong excess of maternal deaths in the March-May 2021 trimester, in general, the rate of excess deaths in previous trimesters was modestly higher than expected or there were no significant differences between observed and expected values in the Southeast, South, and Central-West regions, regardless of trimester assessed. The Southeast region of Brazil, for example, despite being the only region with excess mortality in the aggregated analysis regardless of trimester assessed, after data disaggregation only exhibited strong excess in the number of maternal deaths from March to May 2021. Observed values were relatively close to the maximum expected limit in previous trimesters.

Our results raise concern about the magnitude of COVID-19's impact on maternal mortality throughout the country, as there have been different repercussions among socioeconomically diverse populations that exacerbate pre-existent inequalities, as subgroups of lower socioeconomic position have the least access to healthcare [48, 49]. One study showed reductions in the utilization of maternal health services in 37 health facilities in six LMICs during the COVID-19 pandemic [50]. From March to December 2020, initial antenatal care visits fell by 32% and institutional deliveries dropped by almost 16% [50].

Despite considerable variation between the regions and trimesters assessed, our general results for Brazil showed that the COVID-19 epidemic's impact on maternal mortality in the 10-24-year-old age stratum was substantially smaller than in the rest of the age strata assessed, aligning with studies that point to a lower susceptibility to SARS-CoV-2 infection [51] in younger groups, as well as less time in a hospital bed and an ICU [29] or a smaller occurrence of deaths from the disease [51].

Although the variable age was grouped in different ranges in our study (25–36 and 37–49 years) compared to the grouping (30–39 and 40–49 years) used in the evaluation performed in Brazil between 2015 and 2019 [52], there is relative compatibility with respect to the larger concentration of maternal deaths after the age of 30. Excess maternal mortality was especially high in the South region, with an explosive value of 375% in women 37–49 years of age and in the March-May 2021 trimester. This can be explained by several factors, including the region's disproportionate viral dissemination during the second wave (attributable to circulation of the heavily contagious Gamma variant) [49] in relation to the first wave, during which time the Brazilian South was largely spared of the pandemic's direct and indirect effects. Another important factor to consider is that all other things being equal, pregnant and postpartum women in the Brazilian South, who are generally of a higher average fertility age, are more vulnerable than those in the North region of Brazil, for example, which has the lowest relative participation of older women in the composition of the specific fertility rate [53], as late pregnancy is an independent risk factor for obstetric complications [54, 55].

The COVID-19 pandemic's devastating effects on maternal mortality were also indirectly impacted by the Brazilian government's inability to coherently confront the pandemic, which occurred at an uncontrolled pace in Brazil. Health regulations were insufficient, and the disinformation used by politicians and government authorities made it difficult to implement public health measures to mitigate COVID-19 [56]. For example, the use of drugs shown to be clinically inefficacious for COVID-19 was promulgated [57, 58], and the scientific evidence on protective measures such as the use of masks, social distancing, and the effectiveness and safety of vaccines was deliberately ignored [59].

Brazil initiated COVID-19 vaccination in January 2021 slowly and asynchronously [60]. In addition, the delay in the inclusion of pregnant and postpartum women among the priority groups [61] or even the suspension of vaccinations in pregnant and postpartum women with no comorbidities in mid-May, which only resumed in the first half of July [62], shaped the initial months of the COVID-19 epidemic in 2021. This contrasted with the explosive dissemination of the Gamma variant [14]. Therefore, given the exceptionally high excess of maternal deaths in the March-May 2021 trimester, it is possible that if Brazil guaranteed vaccination of pregnant and postpartum women in a timely manner and accelerated vaccination in the rest of the general population, the indirect effect of the COVID-19 epidemic [20, 63] on maternal mortality could have been smaller [64].

Our study has limitations. Although a portion of the analysed data is considered preliminary, specifically the 2021 data, the fact that we have captured data more than 360 days after the end of the study period drastically limits the possibility of a delay in notification, as well as modifications in the causes of deaths. It is expected that any changes in the dates of deaths would have little influence on the interpretation of our results given the trimonthly temporal aggregation. Another possible limitation is the underreporting of maternal deaths, which would lead to underestimation of the excess in maternal deaths due to the restricted nature of surveillance of deaths among women of childbearing age, especially in the North region, as it is known that during epidemic crises [64] this problem tends to amplify, especially in socioeconomically disadvantaged regions [65]. Regarding the disaggregated analyses, these were useful for elucidating mortality patterns by age group and region but proved unclear whenever the

frequency of deaths was low. We also emphasise that due to the reduced number of recorded maternal deaths in some regions and the unreliability of individual-level data, it was not possible to use characteristics such as age groups usually employed to assess risk factors for pregnancy and postpartum complications, race/skin colour and occupation to identify the specific contributions of age group, racial/ethnic, socioeconomic, and context characteristics, or even those of comorbidities, to excess maternal mortality.

As a strength of this study, we highlight the use of the indicator of excess deaths, which aids in both estimating the magnitude of maternal deaths and capturing part of the COVID-19 pandemic's indirect effects [19] in a disadvantaged health context with known social vulnerabilities. We also emphasize that we evaluated the epidemic in five consecutive trimesters, rather than in one moment, including the moments of greatest impact thus far. This permitted us to identify distinct maternal mortality patterns in Brazil's macroregions and by age group, which could be used not only for mitigating the pandemic's effects but also to enhance mortality information systems in vulnerable regions.

Finally, the observed excess in maternal deaths cannot be attributed exclusively to the health crisis but reflects pre-existing social conditions such as socioeconomic inequalities and limited access to maternal health services, especially in the poorest regions of the country, as well as the precarious management of the health crisis. In this context, the increase in maternal deaths identified in this study represents an unacceptable step backwards, worsening inequalities and social injustices in Brazil and compromising the achievement of SDG 3.

## Supporting information

**S1 Table. Description of causes of death among victims of maternal deaths, by each block of chapter XV of the International Statistical Classification of Diseases and Related Health Problems (ICD) 10th Revision, March 2020 to May 2021, Brazil.**
(DOCX)

**S2 Table. Description of the five main causes of death from the blocks containing the most information on maternal causes of death in chapter XV of International Statistical Classification of Diseases and Related Health Problems (ICD) 10th Revision, according five consecutive trimonthly periods, March 2020 to May 2021, Brazil.**
(DOCX)

## Author Contributions

**Conceptualization:** Jesem Orellana.

**Data curation:** Jesem Orellana.

**Formal analysis:** Jesem Orellana.

**Methodology:** Jesem Orellana.

**Project administration:** Jesem Orellana, Nadège Jacques, Daniel Gray Paschoal Leventhal, Lihsieh Marrero, Lina Sofía Morón-Duarte.

**Supervision:** Jesem Orellana, Nadège Jacques, Daniel Gray Paschoal Leventhal, Lihsieh Marrero, Lina Sofía Morón-Duarte.

**Visualization:** Jesem Orellana, Nadège Jacques, Daniel Gray Paschoal Leventhal, Lihsieh Marrero, Lina Sofía Morón-Duarte.

**Writing – original draft:** Nadège Jacques, Daniel Gray Paschoal Leventhal, Lihsieh Marrero, Lina Sofía Morón-Duarte.

**Writing – review & editing:** Jesem Orellana, Nadège Jacques, Daniel Gray Paschoal Leventhal, Lihsieh Marrero, Lina Sofía Morón-Duarte.

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
