## [Decision Letter · Decision Letter 0]

23 Jun 2022

PONE-D-22-12863Excess maternal mortality in Brazil: Regional inequalities and trajectories during the COVID-19 epidemicPLOS ONE

Dear Dr. Orellana,

Thank you for submitting your manuscript to PLOS ONE. After careful consideration, we feel that it has merit but does not fully meet PLOS ONE’s publication criteria as it currently stands. Therefore, we invite you to submit a revised version of the manuscript that addresses the points raised during the review process. Maternal death and excess mortality during the pandemic were evaluated. Analysis of excess mortality has proven useful for evaluating the effects of the pandemic. Could authors explain the meaning of the counterfactual approach in this well-done analysis?

Interactions of age group and period were explored and more detail in these results is expected.

Further exploring pre-eclampsia, eclampsia, and premature birth would allow for better public policies in maternal health.

We look forward to receiving your revised manuscript.

Kind regards,

Rosely Sichieri

Academic Editor

PLOS ONE

Journal Requirements:

"none"

Additional Editor Comments:

Maternal death and excess mortality during the pandemic were evaluated. Analysis of excess mortality has proven useful for evaluating the effects of the pandemic.

Could authors explain the meaning of the counterfactual approach in this well-done analysis?

Interactions of age group and period were explored and more detail in these results is expected.

Further exploring pre-eclampsia, eclampsia, and premature birth would allow for better public policies in maternal health.

Reviewers' comments:

Reviewer's Responses to Questions

**Comments to the Author**

1. Is the manuscript technically sound, and do the data support the conclusions?

Reviewer #1: Yes

Reviewer #2: Yes

2. Has the statistical analysis been performed appropriately and rigorously? 

Reviewer #1: Yes

Reviewer #2: No

3. Have the authors made all data underlying the findings in their manuscript fully available?

Reviewer #1: Yes

Reviewer #2: Yes

4. Is the manuscript presented in an intelligible fashion and written in standard English?

Reviewer #1: Yes

Reviewer #2: Yes

5. Review Comments to the Author

Reviewer #1: The present article reports the results of a study that was conducted to estimate excess maternal deaths in Brazil and its macroregions as well as their trajectories in the first 15 months of the COVID-19 epidemic. The study used aggregate national data from 2015 until 2021. The study findings showed a 64% excess in maternal deaths in Brazil during the COVID-19 pandemic, with regional heterogeneous trajectories. The manuscript is well written and the analysis appears to be robust and well thought out to achieve the proposed objective. The results are important for the Brazilian and world scenario, as they point out how the Covid pandemic may have an impact on the excess of maternal deaths, especially in low and middle countries, where these deaths are an important public health problem, with a greater burden on the most vulnerable women.

The authors may wish to consider the following comments?

In the discussion, “limited access to family planning and safe abortion services is pointed out as a contributor to the increase in preventable maternal deaths”. The study does not present an analysis by the causes of death to support these hypotheses. We know that eclampsia and preeclampsia have been associated to covid-19, as is also discussed in the article. In this sense, the results would benefit from a brief description of the causes of death during the pandemic period to better discuss the findings that could be placed in complementary material, for example.

Regarding the limitation "The analyzed data are considered preliminary, making it possible for deaths to become reclassified as maternal and for their dates to change..." - It would be important to discuss the direction in which this data gap might affect the results. As the data are preliminary, it is expected that they are underreported, with the excess of death perhaps being greater during the pandemic period. It would be interesting to further discuss these directions and their implications for the study.

Considering the nature of the data, the current state of surveillance of the death of women of childbearing age should also be discussed, especially in the period of the pandemic. should be discussed in the limitations section.

Reviewer #2: Introduction

• The introduction is short and well writhed. However, I suggest describing better the context of the pandemic that supports the hypothesis.

• Maybe, would be interesting, to tell a little about maternal mortality in Brazil. The tragedy involving maternal mortality in Brazil is not exclusively explained by covid-19, though, was aggravated by the pandemic.

Methods

• I suggest describing the study design according to Morgenstern, 1995.

https://www.annualreviews.org/doi/epdf/10.1146/annurev.pu.16.050195.000425

• The number of births could vary considering the period of March 2015 – February 2020 in comparison with 2020 and 2021 (even adjusted for the same period). Then, I think will be safer to calculate the excess mortality by maternal mortality ratio and not by the absolute number of death. To reinforce this idea, there is strong evidence that the number of births decreases in Brazil considering SINASC in 2020.

(https://www.cnnbrasil.com.br/nacional/com-pandemia-numero-de-nascimentos-no-pais-em-2020-e-o-menor-em-26-anos/)

• I understand that the manuscript aims to calculate the excess of maternal mortality. However, could be more interesting, also, analyzed the maternal mortality ratio by region. The results only inform that have excess mortality. But what is the severity of the situation? How was it before the pandemic and how much worse has it become? If the authors conduct this analysis, the manuscript adds value.

• Why was used this classification age (10-24; 25-36 and 37-49)? In maternal mortality it is very common to use 10-19 years– adolescents; 20-35 – adults; and 35 and more – women with increased risk of complications.

Results

• The figures need better definition in legends

Discussion

The discussion is very good. But, maybe needs to suit the new results.

In discussion, some topics could complement to explain the increase in maternal mortality:

1) propagating ineffective drugs and delaying vaccine purchases

2) misinformation about vaccines and the use of masks

3) requirement of a medical prescription for the vaccination of pregnant and postpartum women

4) delay in the approval of vaccination for pregnant and postpartum adolescents

5) delay in the approval of a law on the work remote/license of pregnant women and guarantee of income for pregnant women in informal work situations

6. PLOS authors have the option to publish the peer review history of their article (what does this mean?). If published, this will include your full peer review and any attached files.

Reviewer #1: No

Reviewer #2: **Yes: **Tatiana Henriques Liete

---

## [Author Response · Author response to Decision Letter 0]

5 Aug 2022

Reviewers' comments:

Reviewer's Responses to Questions

Comments to the Author

Review Comments to the Author

Reviewer #1: The present article reports the results of a study that was conducted to estimate excess maternal deaths in Brazil and its macroregions as well as their trajectories in the first 15 months of the COVID-19 epidemic. The study used aggregate national data from 2015 until 2021. The study findings showed a 64% excess in maternal deaths in Brazil during the COVID-19 pandemic, with regional heterogeneous trajectories. The manuscript is well written and the analysis appears to be robust and well thought out to achieve the proposed objective. The results are important for the Brazilian and world scenario, as they point out how the Covid pandemic may have an impact on the excess of maternal deaths, especially in low and middle countries, where these deaths are an important public health problem, with a greater burden on the most vulnerable women.

The authors may wish to consider the following comments?

In the discussion, “limited access to family planning and safe abortion services is pointed out as a contributor to the increase in preventable maternal deaths”. The study does not present an analysis by the causes of death to support these hypotheses. We know that eclampsia and preeclampsia have been associated to covid-19, as is also discussed in the article. In this sense, the results would benefit from a brief description of the causes of death during the pandemic period to better discuss the findings that could be placed in complementary material, for example.

RESPONSE: We thank you for your comments and suggestions. We have provided a brief description of the causes of the death during the pandemic period in Tables S1 and S2. We found that the increases in maternal mortality due to abortion were unclear, thus we removed this point from the Discussion and opted to give a more in-depth exploration of some of the other reasons for high excess maternal mortality, such as pre-eclampsia and eclampsia.

Regarding the limitation "The analyzed data are considered preliminary, making it possible for deaths to become reclassified as maternal and for their dates to change..." - It would be important to discuss the direction in which this data gap might affect the results. As the data are preliminary, it is expected that they are underreported, with the excess of death perhaps being greater during the pandemic period. It would be interesting to further discuss these directions and their implications for the study.

RESPONSE: We appreciate and agree with the reviewer’s point on the status of the previous dataset. The reviewed manuscript uses an updated dataset, including the finalized 2020 data. The following sentences were revised in the Materials and Methods and Discussion sections:

Materials and Methods (lines 85-91):

“Data from 2015 to 2020 [22] are finalized. Those from 2021 were made available preliminarily, up to December 2021. The State Health Departments are given 60 days following the end of the month of each death’s occurrence to transfer their data to SIM’s national platform, though this could possibly take longer [23]. To guarantee reliable coverage to the partial 2021 analyses, deaths occurring until May 2021 were included. Therefore, our analyses refer to a period greater than 360 days following their occurrence with respect to the date the May 2021 deaths data were made available preliminarily, reducing the possibility of a delay in notification [24].”

Discussion (lines 318-321):

“Although a portion of the analysed data is considered preliminary, specifically the 2021 data, the fact that we have captured data more than 360 days after the end of the study period drastically limits the possibility of a delay in notification, as well as modifications in the causes of deaths”

Considering the nature of the data, the current state of surveillance of the death of women of childbearing age should also be discussed, especially in the period of the pandemic. should be discussed in the limitations section.

RESPONSE: We thank the reviewer for their point on the state of the surveillance of deaths during the pandemic. We have revised the parts pertaining to the study limitations in the Discussion section as follows (lines 323-327):

“Another possible limitation is the underreporting of maternal deaths, which would lead to underestimation of the excess in maternal deaths due to the restricted nature of surveillance of deaths among women of childbearing age, especially in the North region, as it is known that during epidemic crises [64] this problem tends to amplify, especially in socioeconomically disadvantaged regions [65].”

 

Reviewer #2: Introduction

• The introduction is short and well writhed. However, I suggest describing better the context of the pandemic that supports the hypothesis.

• Maybe, would be interesting, to tell a little about maternal mortality in Brazil. The tragedy involving maternal mortality in Brazil is not exclusively explained by covid-19, though, was aggravated by the pandemic.

RESPONSE: Thank you very much for your comments and suggestions. Regarding your first point, we prioritized in-depth discussions of the Brazilian pandemic context in lines 300-307 of the Discussion section (below). 

"The COVID-19 pandemic’s devastating effects on maternal mortality were also indirectly impacted by the Brazilian government’s inability to coherently confront the pandemic, which occurred at an uncontrolled pace in Brazil. Health regulations were insufficient, and the disinformation used by politicians and government authorities made it difficult to implement public health measures to mitigate COVID-19 [56]. For example, the use of drugs shown to be clinically inefficacious for COVID-19 was promulgated [57,58], and the scientific evidence on protective measures such as the use of masks, social distancing, and the effectiveness and safety of vaccines was deliberately ignored [59].”

With respect to the second point, we included in the Discussion a more in-depth exploration of the role of obstetric complications such as pre-eclampsia and eclampsia in the rise in maternal mortality (lines 220-226):

“In Brazil, vulnerabilities in obstetric care including difficulties accessing antenatal and specialized care services and a dearth of qualified professionals to handle complications in pregnancy, childbirth, and the postpartum period worsened during the COVID-19 pandemic [34]. The overwhelming burden on health services in the country compromised antenatal care, contributing to the increase in maternal deaths by causes that could have been avoided through adequate pregnancy care, such as pre-eclampsia and eclampsia. It can be observed that the increase in deaths by these causes coincided with the peaks of COVID-19 in the country (Table S2).”

Methods

• I suggest describing the study design according to Morgenstern, 1995.

https://www.annualreviews.org/doi/epdf/10.1146/annurev.pu.16.050195.000425

RESPONSE: Thank you very much for your suggestion. We believe this study used an “exploratory mixed ecologic design” (Morgenstern, 1995) by “combin[ing] the basic features of the multiple-group study and the time trend study.” We performed a time trend analysis, then a prediction ("predict trends in the disease rate for multiple populations") and assumed a context in which the pandemic did not occur to estimate expected maternal deaths (“Assuming the amount of sunlight in the regions has not changed differentially over the study period, we might expect the cohort effect described above to be stronger for sunnier regions").

We described our study accordingly in the Materials and Methods section (lines 77-79): “This exploratory mixed ecologic study [18] employed the counterfactual approach [19] using data from death certificates in the Mortality Information System (SIM) of the Ministry of Health of Brazil during the period of March 2015 to May 2021.”

• The number of births could vary considering the period of March 2015 – February 2020 in comparison with 2020 and 2021 (even adjusted for the same period). Then, I think will be safer to calculate the excess mortality by maternal mortality ratio and not by the absolute number of death. To reinforce this idea, there is strong evidence that the number of births decreases in Brazil considering SINASC in 2020.

(https://www.cnnbrasil.com.br/nacional/com-pandemia-numero-de-nascimentos-no-pais-em-2020-e-o-menor-em-26-anos/)

• I understand that the manuscript aims to calculate the excess of maternal mortality. However, could be more interesting, also, analyzed the maternal mortality ratio by region. The results only inform that have excess mortality. But what is the severity of the situation? How was it before the pandemic and how much worse has it become? If the authors conduct this analysis, the manuscript adds value.

RESPONSE: We appreciate with the reviewer’s suggestion, but we chose to maintain the present approach since our manuscript has a clear goal and a substantial scope, especially after the reviewed version.

• Why was used this classification age (10-24; 25-36 and 37-49)? In maternal mortality it is very common to use 10-19 years– adolescents; 20-35 – adults; and 35 and more – women with increased risk of complications.

RESPONSE: 

Thank you for your comments. We agree that those age groups are useful for distinguishing women of childbearing age by risk of pregnancy and postpartum complications. The following excerpt from the Material and Methods section offers our rationale for using the age groups employed (lines 130-133):

“Due to the reduced number of maternal deaths that occurred before the age of 20 and the reduced accuracy in our maternal death estimates, age groups were aggregated in the following brackets: 10-24 years, 25-36 years, and 37-49 years”

Results

• The figures need better definition in legends

RESPONSE: We greatly appreciate your suggestion. We have taken care to ensure all figures and legends are clearly legible. 

Discussion

The discussion is very good. But, maybe needs to suit the new results.

In discussion, some topics could complement to explain the increase in maternal mortality:

1) propagating ineffective drugs and delaying vaccine purchases

2) misinformation about vaccines and the use of masks

3) requirement of a medical prescription for the vaccination of pregnant and postpartum women

4) delay in the approval of vaccination for pregnant and postpartum adolescents

5) delay in the approval of a law on the work remote/license of pregnant women and guarantee of income for pregnant women in informal work situations

RESPONSE: We greatly appreciate your comments and generous suggestions. In lines 300-307 of the Discussion section, we addressed these issues in greater detail:

“The COVID-19 pandemic’s devastating effects on maternal mortality were also indirectly impacted by the Brazilian government’s inability to coherently confront the pandemic, which occurred at an uncontrolled pace in Brazil. Health regulations were insufficient, and the disinformation used by politicians and government authorities made it difficult to implement public health measures to mitigate COVID-19 [56]. For example, the use of drugs shown to be clinically inefficacious for COVID-19 was promulgated [57,58], and the scientific evidence on protective measures such as the use of masks, social distancing, and the effectiveness and safety of vaccines was deliberately ignored [59].”

---

## [Editor Report · Decision Letter 1]

14 Sep 2022

Excess maternal mortality in Brazil: Regional inequalities and trajectories during the COVID-19 epidemic

PONE-D-22-12863R1

Dear Dr. Orellana,

We’re pleased to inform you that your manuscript has been judged scientifically suitable for publication and will be formally accepted for publication once it meets all outstanding technical requirements. I am sorry for the delay.

Kind regards,

Rosely Sichieri

Academic Editor

PLOS ONE

---

## [Editor Report · Acceptance letter]

12 Oct 2022

PONE-D-22-12863R1 

Excess maternal mortality in Brazil: Regional inequalities and trajectories during the COVID-19 epidemic 

Dear Dr. Orellana:

I'm pleased to inform you that your manuscript has been deemed suitable for publication in PLOS ONE. Congratulations! Your manuscript is now with our production department. 

Kind regards, 

on behalf of

Dr. Rosely Sichieri 

Academic Editor

PLOS ONE